# The Mediterranean Diet Protects Renal Function in Older Adults: A Prospective Cohort Study

**DOI:** 10.3390/nu14030432

**Published:** 2022-01-19

**Authors:** Ana Bayán-Bravo, Jose Ramón Banegas, Carolina Donat-Vargas, Helena Sandoval-Insausti, Manuel Gorostidi, Fernando Rodríguez-Artalejo, Pilar Guallar-Castillón

**Affiliations:** 1Department of Preventive Medicine and Public Health, School of Medicine, Universidad Autónoma de Madrid-IdiPaz, CIBERESP (CIBER of Epidemiology and Public Health), 28029 Madrid, Spain; a.bayan.bravo@gmail.com (A.B.-B.); joseramon.banegas@uam.es (J.R.B.); cdonatvargas@gmail.com (C.D.-V.); fernando.artalejo@uam.es (F.R.-A.); 2Clinical Nutrition and Dietetics Unit, Department of Endocrinology and Nutrition, 12 de Octubre Hospital, 28041 Madrid, Spain; 3IMDEA-Food Institute, CEI UAM+CSIC, 28049 Madrid, Spain; 4Department of Nutrition, Harvard T.H. Chan School of Public Health, Boston, MA 02115, USA; helenagabar@gmail.com; 5Department of Nephrology, Hospital Universitario Central de Asturias. Red de Investigación Renal (RedinRen), 33011 Oviedo, Spain; manuel.gorostidi@gmail.com

**Keywords:** Mediterranean diet, *a posteriori* patterns, renal function decline, older adults

## Abstract

Background: Chronic kidney disease entails a high disease burden that is progressively increasing due to population aging. However, evidence on the effect of the Mediterranean diet on renal function is limited, in particular among older adults in Mediterranean countries. Methods: Prospective cohort study with 975 community-dwelling adults aged ≥ 60 recruited during 2008–2010 in Spain and followed up to 2015. At baseline, food consumption was obtained using a validated dietary history. Two Mediterranean dietary patterns were used: (i) An a priori-defined pattern, the Mediterranean Diet Adherence Screener (MEDAS score: low adherence: 0–5 points; moderate: 6–8 points; high: 9–14 points); (ii) An *a posteriori* Mediterranean-like dietary pattern, based on 36 food groups, which was generated using factor analysis. Renal function decline was calculated as an estimated glomerular filtration rate (eGFR) decrease ≥1 mL/min/1.73 m^2^ per year of follow-up. Results: A total of 104 cases of renal function decline occurred. Compared with participants with a low MEDAS adherence, the multivariable-adjusted odds ratios (95% confident interval) for renal function decline risk were 0.63 (0.38–1.03) for moderate adherence, and 0.52 (0.29–0.95) for high adherence (*p*-trend: 0.015). Multivariable-adjusted odds ratios (95% confidence interval) for renal function decline risk according to increasing quartiles of the adherence to the *a posteriori* Mediterranean-like dietary pattern were 1.00, 0.67 (0.38–1.20), 0.65 (0.35–1.19), and 0.47 (0.23–0.96) (*p*-trend: 0.042). Conclusion: A higher adherence to a Mediterranean diet was associated with a lower risk of renal function decline in older adults, suggesting benefits to health of this dietary pattern in Mediterranean countries.

## 1. Introduction

Chronic kidney disease (CKD) is a major public health concern worldwide. CKD is related to an increase in all-cause and cardiovascular mortality, and renal function impairment involves a substantial economic and social burden on the general population. In Spain, CKD affects one in seven adults with a prevalence that is similar to that of the United States [1]. Moreover, health-related quality of life diminishes from the early stages of CKD and progressively worsens across the stages of this disease [2,3].

Several dietary strategies have been suggested to slow age-related kidney function impairment [2]. Two meta-analyses of cohort studies showed that healthy dietary patterns were associated with a reduced incidence of CKD in primary prevention, and with a lower mortality risk in patients who already have CKD [3,4]. However, in a sub-analysis of the Nurses’ Health Study conducted with 3071 American women (mean age 67), the adherence to a prudent dietary pattern (with high intake of fruits, vegetables, legumes, fish, poultry and whole grains) was not linked to renal function decline [5].

It is uncertain whether a well-known healthy dietary pattern, such as the Mediterranean diet, is beneficial or not for the prevention of renal function decline associated with aging. In the Uppsala Longitudinal Study of Adult Men (ULSAM), a population-based cohort of Swedish men aged ≥50, adherence to a Mediterranean diet was associated with a reduced incidence of CKD, and a lower mortality among those with CKD [6]. Likewise, some prospective cohort studies in the USA have shown a beneficial association between the adherence to the Mediterranean diet and the risk of CKD in free living populations [7,8]. On the contrary, in a sub-cohort analyses of the PREDIMED Study (Spanish community-dwelling men aged 55–80 and women aged 60–80, at a high cardiovascular risk), after 1 year of follow-up, those participants who followed a Mediterranean diet did not have a beneficial effect on kidney function decline, when compared with participants who received advice on a low-fat diet [9].

Given the limited and inconsistent evidence of the relationship between the Mediterranean diet and renal function decline, in particular among older adults, we examined the prospective association between two Mediterranean dietary patterns and the risk of renal function decline in older men and women from Spain. To assess the robustness of the association we used a well-established “a priori” Mediterranean dietary pattern as well as an *a posteriori* Mediterranean dietary pattern identified through factor analysis.

## 2. Methods

### 2.1. Study Design and Participants

Data were obtained from the Seniors-ENRICA cohort, which included 2519 community-dwelling Spanish participants aged 60 and older. The cohort comes from the ENRICA study, a representative cohort of the non-institutionalized adult population established in 2008–2010 [10]. Baseline information was collected in three stages: firstly, a phone interview to obtain information on health status, lifestyle, and morbidity; secondly, a home visit to collect blood and urine samples; and thirdly, a second home visit to perform a physical examination and to record habitual diet. After 6.5 years of follow-up, 616 participants were lost, 82 died and we followed 1821 participants until 2015 when using the same procedures, a new phone interview and home visit were conducted.

Of these 1821 participants, serum creatinine (SCr) mesurements at baseline and at the end of the follow-up were available on 1468 participants. Of these, 493 were excluded for having missing values in potential confounders. Thus, the analyses were conducted with 975 participants.

Participants gave written informed consent, and the study was approved by the Clinical Research Ethics Committee of La Paz University Hospital in Madrid.

### 2.2. Study Variables

#### 2.2.1. Diet

At baseline, information on diet was collected through a validated computerized face-to-face dietary history (HD-ENRICA) [11], developed based on that used in the EPIC-Spain (European Prospective Investigation into Cancer and Nutrition) cohort study. Participants were asked about the food consumed in a typical week representing the preceding year. Collected information included all foods consumed at least once every 15 days. The HD-ENRICA allows you to record information on 860 foods using 120 sets of photographs to help estimating their portion sizes. Diet was collected by trained and certified interviewers. Energy intake was estimated using standard food composition tables from Spain [12,13].

The adherence to the Mediterranean diet was assessed with the Mediterranean Diet Adherence Screener index (MEDAS) [14], developed to evaluate the adherence to the Mediterranean diet in the PREDIMED study [15]. This score includes 1 item on olive oil use for cooking, that is characteristic of the traditional Mediterranean diet in Spain (Do you use olive oil as the principal source of fat for cooking?); 11 items related to the intake frequency of some food groups (e.g., How many times do you consume nuts per week? How many sugar-sweetened beverages do you consume per day?). Of these 11 food groups, 7 scored positively (intake of olive oil, vegetables, fruit, wine, legumes, fish, and nuts), and 4 scored negatively (intake of red meat, butter, sweetened beverages, and commercial pastries); and 2 additional items related to food intake habits: Do you prefer to eat chicken, turkey or rabbit instead of beef, pork, hamburgers, or sausages? How many times per week do you consume boiled vegetables, pasta, rice, or other dishes with a sauce of tomato, garlic, onion, or leeks sautéed in olive oil? Each question was scored 0 or 1, and the MEDAS score ranges from 0 to 14.

To identify *a posteriori* dietary patterns, the 860 foods were categorized into 36 different groups according to similarities in their nutrient profile. Next, we applied factor analysis (principal components analysis) to these food groups to generate various independent dietary patterns (factors) made up of foods with a high degree of correlation [14]. The factors were rotated by orthogonal transformation (Varimax rotation) to facilitate their interpretation, and we kept those factors with an eigenvalue ≥1.5 on the screen test [16]. Two dietary patterns were identified, and loading factors were obtained for each food group to identify those that were more closely correlated with each dietary pattern.

The first pattern, rich in refined bread, whole dairy products, red and processed meat, and low in whole grains, fruit, low-fat dairy, and vegetables, was called the *a posteriori* Western-like dietary pattern. The second pattern, rich in olive oil, vegetables, potatoes, legumes, blue fish, pasta, and white meat, was called the *a posteriori* Mediterranean-like dietary pattern. Loading factors for each pattern in this cohort have been published elsewhere [15]. Finally, for both patterns each participant received a score that was calculated as the sum of the consumption of each food group weighted by the corresponding loading factor. A higher score indicated a greater adherence to the respective dietary pattern.

#### 2.2.2. Serum Creatinine and Estimated Glomerular Filtration Rate

At baseline and at the end of follow-up, a 12-h fasting blood and a spot urine sample were obtained during the home visits. Laboratory determinations were performed centrally at the Center of Biological Diagnosis of the Hospital Clínic (Barcelona) using standard procedures and appropriate quality controls [10]. SCr was determined by the Jaffé’s kinetic reaction with alkaline picrate. The eGFR at baseline and in 2015 was calculated with the Chronic Kidney Disease Epidemiology Collaboration (CKD-EPI) Equation [17]. Renal function decline was considered when the decrease in the eGFR during follow-up was greater than 1 mL/min/1.73 m^2^ per year.

#### 2.2.3. Other Variables

At baseline, study participants reported data on socio-demographic and lifestyle variables, including sex, age, educational level (primary, secondary, and university), smoking status (never, former, or current smoker), and alcohol consumption (habitual, occasional, non-drinker, or ex-drinker). Leisure-time physical activity and physical activity in the household were obtained at baseline with the questionnaire developed by the EPIC group and were expressed in metabolic equivalents (MET)-h/week [18]. Baseline information on the time spent watching TV (in h/week) and night sleeping time (in h/day) were also obtained. Study participants also reported the following physician-diagnosed diseases: cardiovascular diseases (including ischemic heart disease, stroke, and heart failure), chronic bronchitis or asthma, cancer, and osteo-muscular disease (osteoarthritis and arthritis). Weight, height, and waist circumference were measured under standardized conditions; body mass index (BMI) was calculated as weight in kg divided by height in m squared. Blood glucose was measured by the glucose oxidase method after 12-h fasting. Blood pressure was measured under standardized conditions with validated automatic sphygmomanometers. Serum high-density lipoprotein cholesterol (HDL-c) and triglyceridemia were measured by the direct method using elimination/catalase and the glycerol phosphate oxidase method. Cardiovascular risk factors were included in the models as dichotomous variables according to the categories used to define the metabolic syndrome [19] as the following: high glucose: fasting glucose ≥ l00 mg/dL; high blood pressure: systolic blood pressure ≥ 130 mmHg and/or diastolic blood pressure ≥ 85 mmHg; abdominal obesity: waist circumference ≥ 102 cm in men and ≥88 cm in women, low HDL-c: HDL-c < 40 mg/dL in men or < 50 mg/dL in women; high triglicerides: serum triglicerides ≥ 150 mg/dL. Muscular mass was measured in 2015 using bioelectrical impedance (Tanita SC240-Ma; Tanita Corporation; Tokio; Japan).

### 2.3. Statistical Analysis

The association between the adherence to the MEDAS score and the risk of renal function decline was summarized with odds ratios (OR) and their 95% confidence interval obtained from logistic regression. Participants were categorized into three categories according to their adherence to the MEDAS score as low (0–5 points), moderate (6–8 points), and high (9–14 points), and the first category (lowest adherence) was used as reference. Four sequential logistic models were built. The first model was adjusted for sex, age, educational level, and total energy intake. The second model was further adjusted for smoking status, alcohol consumption, leisure-time physical activity, household physical activity, time spent watching TV, night sleeping time (h/day), and chronic morbidity. The third model was additionally adjusted for BMI and prevalent cardiovascular risk factors (including high glucose, high blood pressure, abdominal obesity, low HDL-c, and high triglycerides). Model 3 was considered the main model. Model 4 was additionally adjusted for incident components of the metabolic syndrome such as a BMI increase during follow-up (yes/no), cardiovascular risk factors in 2015 as dichotomous variables (including high serum glucose, high blood pressure, high waist circumference, low HDL-c, and high triglycerides), and muscle mass in 2015. This model was not the main model because some of the variables in model 4 could also be mediators of the association. Lastly, in model 5, the two *a posteriori* dietary patterns were mutually adjusted.

Statistical significance was set at two-sided *p* < 0.05. The analyses were performed with Stata/SE, version 13.1 (StataCorp, College Station, TX, USA).

Clinical Trial Registry: ClinicalTrials.gov Identifier: NCT02804672.

## 3. Results

After a 6.5-year mean follow-up, 104 (10.7%) subjects experienced renal function decline. Among those with renal function decline, the increase in SCr levels was from a mean of 0.93 to 1.15 mg/dL (absolute difference: 0.22 mg/dL percentage increase: 23.7%), and the decrease in eGFR from a mean of 76.5 to 61.4 mL/min/1.73 m^2^ (absolute difference: 15.1 mL/min/1.73 m^2^; percentage decrease: 19.7%).

At baseline, the mean age of the 975 participants was 67.4 ± 5.43 and 48.6% were women. Participants with renal function decline (when compared with those who did not) were less often women, were older, more frequently current smokers, did less household physical activity, spent more time watching TV, had more frequently cardiovascular diseases as well as a worse cardiovascular risk factor profile (low HDL-c and high triglycerides more frequently) (Table 1).

Compared to individuals with a low adherence to the MEDAS index, those with a moderate adherence had a risk of renal function decline of OR: 0.63 (95% CI: 0.38–1.03), and those with a high adherence had a significantly lower risk OR: 0.52 (95% CI: 0.29–0.95); *p*-trend 0.015 (Table 2, model 3). Model 4 produced similar results.

When each component of the MEDAS index was analyzed individually, those that contributed most to the beneficial association of the Mediterranean pattern on renal function decline were vegetable consumption, low intake of sweetened beverages, and preference for white meat instead of beef, pork, hamburgers, or sausages (data not shown).

When the *a posteriori* Western-like dietary pattern was considered, no association was found with renal function decline (Table 3, model 3). On the contrary, a significant and inverse association was found for the Mediterranean-like dietary pattern. The ORs for renal function decline across increasing quartiles of this pattern were 1.00, 0.67 (95% CI: 0.38–1.20), 0.65 (0.35–1.19), and 0.47 (0.23–0.96); *p*-trend: 0.042 (Table 3, model 3). Similar results were found after controlling for some potential mediators and for the western-like dietary pattern (Table 3, model 5).

## 4. Discussion

In this population-based cohort of older adults, a high adherence to the Mediterranean dietary pattern using the MEDAS-index was independently associated with a significant 48% lower risk of renal function decline after 6.5 years of follow-up. Similar results were obtained when examined the adherence to the Mediterranean-like dietary pattern defined *a posteriori*. Our results extent the evidence favoring the promotion of the Mediterranean Diet to prevent the deterioration of the renal function among older men and women who live in the community.

A recent meta-analysis conducted with eight observational studies confirmed that healthy dietary patterns are linked to a reduced incidence of CKD and incidence of albuminuria, but the association did not reach significance for eGFR decline [3]. A previous meta-analysis had already shown the association of following a healthy dietary patterns and lower mortality in participants with kidney disease [4]. These healthy patterns were characterized by a high consumption of fruit and vegetables, fish, legumes, whole grains, and fiber, and a low consumption of red meat, sodium, and refined sugars. One of the meta-analyzed studies (the ULSAM study) showed that the adherence to the Mediterranean diet was linked to a lower incidence of CKD in a population-based cohort of older men, as well as to a reduced mortality risk in patients with manifest CKD [6]. Moreover, in the Northern Manhattan Study (participants aged ≥ 55), the adherence to a Mediterranean diet was linked to a reduced CKD risk, as well as a decreased renal function decline in this primary prevention multi-ethnical cohort after 6.9 years of follow-up [7]. In addition, analyses conducted in another multi-ethnical prospective cohort, the Atherosclerosis Risk in Communities Study (ARIC Study) among American participants aged 45–64, and after a long follow-up period, showed similar results [8]. A higher adherence to a Mediterranean diet during middle age was also associated with a lower risk of CKD in later life. Thus, in general, prospective studies conducted in the USA (following westernized dietary habits), the adherence to the Mediterranean diet showed clearer benefits on kidney function.

In contrast to this, the PREDIMED study [9] conducted in Spain, was unable to prove the beneficial effect of the Mediterranean diet on kidney function after 1-year follow-up when compared with participants who received advice on a low-fat diet. This might be because the follow-up period was not long enough in order for the disease to develop, especially considering that all participant received counseling in order to follow a healthy diet.

Certain biological mechanisms may account for the relationship observed between the Mediterranean dietary pattern and renal function decline. Thus, the Mediterranean diet has been inversely linked to low-grade inflammation, oxidative stress, as well as endothelial impairment [20,21,22,23] This dietary pattern diminishes the levels of classic inflammatory biomarkers, such as C-reactive protein and interlukin-6 as well as more recent inflammatory markers like platelet count [21]. The presence of chronic low-grade inflammation has been established as a major underlying cause for CKD [24]. An increased inflammatory state has also been quantified by imaging in the arterial walls of patients with CKD [25].

Apart from a low-grade inflammatory background, the Mediterranean diet was associated with an improvement in lipid profile, blood pressure, and insulin sensitivity [26,27,28]. The Mediterranean diet also improves endothelial function. Thus, evidence from a systematic review and meta-analysis of randomized controlled trials indicates that the Mediterranean diet improves endothelial function in adults, even at early stages of the atherosclerotic process by improving flow-mediated dilation, and this improvement is independent of the participants’ health status [29].

The source of dietary proteins intake is also relevant for kidney function. Several studies indicated that diets rich in fruit, vegetables, fish, cereals, whole grains, and polyunsaturated fatty acids but low in saturated fatty acids and red meat were beneficial for CKD patients [30,31,32,33,34] In the Mediterranean diet the most important sources of proteins are vegetables, fish, and white meat, which have been linked to lower CKD incidence and improvements in the end-stages of the renal disease [26,35]

Fiber content of the diet also seems to play a pivotal role in renal function decline. Diets rich in fiber (with high fruit and vegetable intake), such as the Mediterranean diet, are also beneficial for primary kidney disease prevention and, once the disease has been diagnosed, for delaying CKD progression as well as its related complications [4,31,33]. Moreover, a diet rich in whole grains and low in dietary red and processed meat is beneficial among patients with impaired kidney function, both reducing inflammation [36] as well as improving the phosphorus homeostasis that is altered in CKD patients [33,37,38].

The Mediterranean diet also has a high content of omega-3 polyunsaturated fatty acids (that are mainly obtained from fatty fish consumption) which can act again, through their anti-inflammatory and anti-thrombotic properties [39], while being low in saturated fatty acids, that are pro-inflammatory nutrients [40].

All the above makes the Mediterranean diet the diet of choice for patients with established CKD [26], as well as in the general population, to counteract the deleterious effect of age on decreasing renal function. Our results extend these benefits to the Mediterranean countries.

The study’s strengths include its prospective design as well as its dietary data collection, which was performed using a validated dietary history by trained and certified interviewers, including information on a large number or food items. Additionally, it is a population-based study using an unselected sample of community-dwelling older men and women. In addition, the analyses were controlled for a relatively large number of potential confounders including incident cardiovascular risk factors and muscle mass. Moreover, the Mediterranean diet was assessed using two different methods including factor analysis, which is a valid tool to define dietary patterns without making previous assumptions on the correlation of the data. The concordance between the two methods strengthens our findings.

Some limitations should be noted. Diet was self-reported and, as in most nutritional epidemiological studies, a certain recall bias cannot be dismissed. In addition, some residual confounding cannot be ruled out, although we did adjust for many potential confounders. In addition, even though the renal function decline we observed in the follow-up was not extremely high, it should be kept in mind that in population studies there is an independent and continuous association of SCr increase and eGFR decrease with mortality risk, even when eGFR is within the normal range (from 60 to 74 mL/min/1.73 m^2^) [41,42]. Thus, almost any decline in renal function might be meaningful and responsible for a significant proportion of fatal and non-fatal events [42], augmented by the high frequency of older adults whose renal function declines naturally over time. The use of the Jaffè method in the measurement of creatinine may lead to an overestimation (eGFR underestimation [43]. However, this error might introduce a non-differential bias, and would not affect the found associations. Lastly, we only had a single SCr measurement both, at baseline and at follow-up, which could also lead to non-differential misclassifications that bias the results to the null.

In conclusion, in this prospective study of community-dwelling older adults in Spain, a high adherence to a normative Mediterranean dietary index or to an *a posteriori* Mediterranean-like dietary pattern was protective against renal function decline. These findings extend the observational evidence on the beneficial effects of the Mediterranean diet among older adults from Mediterranean countries. It also provides us with clear dietary recommendation for primary prevention of renal function decline. Finally, further clinical trials are needed to confirm the efficacy of the adherence to a Mediterranean diet in preventing renal function decline among unselected community-dwelling participants.

## Figures and Tables

**Table 1 nutrients-14-00432-t001:** Baseline characteristics of the Seniors-ENRICA participants according to renal function decline after 6.5 years of follow-up (2008/2010–2015) *N* = 975.

	Total	No Renal Function Decline	Renal Function Decline	*p*-Value
N (%)	975 (100)	871 (89.3)	104 (10.7)	
Sex, women, %	48.6	49.7	39.4	0.047
Age, in years, mean (SD)	67.4 (5.43)	67.2 (5.3)	69.0 (6.1)	0.014
eGFR (SD)	77.9 (12.2)	78.2 (12.0)	76.5 (14.2)	0.189
eGFR categories, %				
Category 1 (≥90)	18.2	18.4	16.4	0.027
Category 2 (≥60–89)	74.2	74.4	72.1	
Category 3a (≥45–59)	6.8	6.5	8.7	
Category 3b (≥30–44)	0.8	0.7	1.9	
Category 4 (≥15–29)	0.1	0.0	1.0	
Category 5 (<15)	0	0	0	
Serum creatinine	0.90 (0.18)	0.89 (0.17)	0.93 (0.20)	0.048
Educational level, %				
Primary	48.0	47.3	53.9	0.339
Secondary	26.7	27.3	21.1	
University	25.3	25.4	25.0	
Energy intake in kcal, mean (SD)	2046 (571)	2054 (565)	1974 (617)	0.182
Smoking status, %				
Never smoker	56.6	57.2	51.9	0.032
Former smoker	31.9	32.3	28.9	
Current smoker	11.5	10.6	19.2	
Alcohol consumption, %				
Habitual drinker	47.5	48.5	39.4	0.244
Occasional drinker	21.4	21.4	22.1	
Non-drinker	23.6	22.7	30.8	
Ex-drinker	7.5	7.5	7.7	
Leisure-time physical activity, MET-h/week, mean (SD)	22.6 (15.7)	22.9 (15.6)	21.0 (16.2)	0.075
Household physical activity, MET-h/week, mean (SD)	36.8 (31.6)	37.5 (31.4)	31.0 (32.5)	0.049
Time spent watching TV in h/week, mean (SD)	17.4 (11.1)	17.0 (10.8)	20.1 (13.1)	0.008
Night sleeping time in h/day, mean (SD)	7.14 (1.43)	7.13 (1.43)	7.20 (1.49)	0.664
Cardiovascular disease, %	1.23	0.92	3.85	0.010
Bronchitis or asthma, %	7.7	7.8	6.7	0.697
Cancer, %	2.4	2.4	1.9	0.096
Osteo-muscular disease, %	46.7	46.0	51.9	0.256
BMI at baseline in kg/m^2^, mean (SD)	28.5 (4.16)	28.5 (4.1)	29.1 (4.6)	0.119
Metabolic syndrome components at baseline				
Glucose ≥ 100, %	43.6	42.6	51.2	0.070
Blood pressure ≥ 130/85, %	80.7	80.6	81.7	0.782
WC ≥ 102 men, ≥ 88 women, %	56.3	55.3	64.4	0.078
HDL-c < 40 men, < 50 women, %	22.8	21.8	30.8	0.040
Triglycerides ≥ 150, %	20.9	19.9	29.8	0.018
BMI in 2015 in kg/m^2^, mean (SD)	27.9 (4.3)	27.9 (4.2)	28.6 (4.6)	0.073
Metabolic syndrome components in 2015				
Glucose ≥ 100, %	41.0	40.0	50.0	0.049
Blood pressure ≥ 130/85, %	66.7	65.8	74.0	0.092
WC ≥ 102 men, ≥ 88 women, %	59.4	58.7	65.4	0.187
HDL-c < 40 men, < 50 women, %	27.9	26.4	40.4	0.003
Triglycerides ≥ 150, %	16.5	15.7	23.1	0.056
Muscle mass in kg, mean (SD)	46.7 (9.4)	46.6 (9.5)	47.9 (9.1)	0.197

SD (Standard Deviation); eGFR: estimated glomerular filtration rate (Stratification according to the Kidney Disease: Improving Global Outcomes KDIGO 2012). BMI: Body Mass Index, WC: waist circumference; HDL-c: High-density lipoprotein cholesterol. *p* values are based on chi-square test for qualitative variables or *t*-test for continuous variables.

**Table 2 nutrients-14-00432-t002:** Association between the adherence to the Mediterranean Diet Adherence Screener (MEDAS) index and the risk of renal function decline after 6.5-years of follow-up in the Seniors-ENRICA study (2008/2010–2015) *N* = 975.

Renal Function Decline	Low Adherence to MEDAS (0–5 Points) OR (95% CI)	Moderate Adherence to MEDAS (6–8 Points) OR (95% CI)	High Adherence to MEDAS (9–14 Points) OR (95% CI)	*p* Trend
Cases/total 104/975	54/414	33/314	17/247	
Model 1	Ref.	0.78 (0.49–1.26)	0.46 (0.26–0.83) †	0.009
Model 2	Ref.	0.67 (0.41–1.09)	0.53 (0.29–0.95) *	0.020
Model 3	Ref.	0.63 (0.38–1.03)	0.52 (0.29–0.95) *	0.015
Model 4	Ref.	0.61 (0.37–1.01)	0.52 (0.28–0.94) *	0.016

* *p* < 0.05; † *p* < 0.01. OR: Odds Ratio. CI: Confidence Interval. Model 1: Adjusted for sex, age, eGFR at baseline, education level (primary, secondary, or university), and total energy intake. Model 2: Additionally adjusted for smoking status (never, former, or current-smoker), alcohol consumption (habitual drinker, occasional drinker, non-drinker, or ex-drinker), leisure-time physical activity (MET-h/week), household physical activity (MET-h/week), occupational physical activity (sedentary, light, moderate/vigorous, or none), time spent watching TV (h/week), night sleeping time (h/day), prevalence of cardiovascular disease, bronchitis or asthma, cancer, and osteo-muscular disease. Model 3: Additionally adjusted for BMI at baseline and prevalent components of the metabolic syndrome (glucose ≥ 100 mg/dL (yes, no), blood pressure ≥ 130/85 mmHg (yes, no), abdominal obesity ≥ 102 cm in men and ≥ 88 cm in women (yes, no), HDL-c < 40 mg/dL in men and < 50 mg/dL in women (yes, no), triglycerides ≥ 150 mg/dL (yes, no). Model 4: Additionally adjusted for an increase in BMI (yes, no) and incident components of the metabolic syndrome developed during 2008/10 to 2015 (glucose ≥ 100 mg/dL (yes, no), blood pressure ≥ 130/85 mmHg (yes, no), abdominal obesity ≥ 102 cm in men and ≥ 88 cm in women (yes, no), HDL-c < 40 mg/dL in men and < 50 mg/dL in women (yes, no), triglycerides ≥ 150 mg/dL (yes, no), and muscular mass (kg).

**Table 3 nutrients-14-00432-t003:** Association between the adherence to *a posteriori* dietary patterns and the risk of renal function decline after 6.5-years of follow-up in the Seniors-ENRICA study (2008/2010–2015) *N* = 975.

		Western-Like Pattern				Mediterranean-Like Pattern				
Renal Function Decline	Q1 OR (95% CI) (Lowest)	Q2 OR (95% CI)	Q3 OR (95% CI)	Q4 OR (95% CI) (Highest)	*p* Linear Trend	Q1 OR (95% CI) (Lowest)	Q2 OR (95% CI)	Q3 OR (95% CI)	Q4 OR (95% CI) (Highest)	*p* Linear Trend
Cases/total 104/975	20/244	31/244	29/244	24/243		37/244	27/243	23/244	17/244	
Model 1	1 Ref.	1.68 (0.91–3.09)	1.61 (0.84–3.07)	1.61 (0.76–3.42)	0.235	1 Ref.	0.68 (0.39–1.18)	0.58 (0.32–1.04)	0.40 (0.20–0.80) †	0.008
Model 2	1 Ref.	1.60 (0.85–3.00)	1.58 (0.81–3.08)	1.42 (0.65–3.09)	0.381	1 Ref.	0.67 (0.38–1.18)	0.63 (0.34–1.17)	0.45 (0.22–0.92) *	0.030
Model 3	1 Ref.	1.59 (0.85–2.98)	1.50 (0.77–2.92)	1.35 (0.61–2.97)	0.474	1 Ref.	0.67 (0.38–1.20)	0.65 (0.35–1.19)	0.47 (0.23–0.96) *	0.042
Model 4	1 Ref.	1.65 (0.87–3.12)	1.54 (0.78–3.05)	1.43 (0.64–3.18)	0.411	1 Ref.	0.70 (0.39–1.24)	0.68 (0.36–1.26)	0.47 (0.23–0.97) *	0.049
Model 5	1 Ref.	1.62 (0.86–3.07)	1.37 (0.68–2.73)	1.12 (0.49–2.57)	0.798	1 Ref.	0.68 (0.38–1.22)	0.66 (0.35–1.25)	0.45 (0.21–0.96) *	0.070

* *p* < 0.05; † *p* < 0.01. OR: Odds Ratio. CI: Confidence Interval. Model 1: Adjusted for sex, age, eGFR at baseline, education level (primary, secondary, or university), and total energy intake. Model 2: Additionally adjusted for smoking status (never, former, or current-smoker), alcohol consumption (habitual drinker, occasional drinker, non-drinker, or ex-drinker), leisure-time physical activity (MET-h/week), household physical activity (MET-h/week), occupational physical activity (sedentary, light, moderate/vigorous, or none), time spent watching TV (h/week), night sleeping time (h/day), prevalence of cardiovascular disease, bronchitis or asthma, cancer, and osteo-muscular disease. Model 3: Additionally adjusted for BMI at baseline and prevalent components of the metabolic syndrome (glucose ≥ 100 mg/dL (yes, no), blood pressure ≥ 130/85 mmHg (yes, no), abdominal obesity ≥ 102 cm in men and ≥ 88 cm in women (yes, no), HDL-c < 40 mg/dL in men and < 50 mg/dL in women (yes, no), triglycerides ≥ 150 mg/dL (yes, no). Model 4: Additionally adjusted for an increase in BMI (yes, no) and incident components of the metabolic syndrome developed during 2008/10 to 2015 (glucose ≥100 mg/dL (yes, no), blood pressure ≥ 130/85 mmHg (yes, no), abdominal obesity ≥ 102 cm in men and ≥ 88 cm in women (yes, no), HDL-c < 40 mg/dL in men and < 50 mg/dL in women (yes, no), triglycerides ≥ 150 mg/dL (yes, no), and muscular mass (kg). Model 5: Additionally adjusted for the other *a posteriori* dietary pattern, as appropriate.

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
