# Peer review of "The Mediterranean Diet Protects Renal Function in Older Adults: A Prospective Cohort Study"

_nutrients, 2022, doi:10.3390/nu14030432_

Round 1

Reviewer 1 Report

This paper reports the results of an interesting study aimed to examine the prospective association between two Mediterranean dietary patterns and the risk of renal function decline in older men and women from Spain. Chronic kidney disease (CKD) is a major public health concern worldwide. For this reason, this study could improve the readers insight about the evidence of the relationship between the Mediterranean diet and renal function declines. This paper reports results on a large community-dwelling adults aged ≥60 (n=975). This is an important strength both because it represents a large effort and statistical solid results and because this is a complex and understudied adults aged ≥60 population.

Minor revision:

- In results section, I do not understand the dexcription made of the sample, 1821 in total, 616 are lost, and 1205 remain. How do you get 975 at the end?

Author Response

Reviewer 1

This paper reports the results of an interesting study aimed to examine the prospective association between two Mediterranean dietary patterns and the risk of renal function decline in older men and women from Spain. Chronic kidney disease (CKD) is a major public health concern worldwide. For this reason, this study could improve the readers insight about the evidence of the relationship between the Mediterranean diet and renal function declines. This paper reports results on a large community-dwelling adults aged ≥60 (n=975). This is an important strength both because it represents a large effort and statistical solid results and because this is a complex and understudied adults aged ≥60 population.

Minor revision:

- In results section, I do not understand the description made of the sample, 1821 in total, 616 are lost, and 1205 remain. How do you get 975 at the end?

Authors: We appreciate this remark. The difference was explained because 493 participants had missing values in potential confounders at baseline or at the end of follow-up. We have rewording the paragraph to make it clearer. New you can see in the text:

“Of these 1,821 participants, serum creatinine (SCr) mesurements at baseline and at the end of the follow-up were available on 1,468 participants. Of these, 493 were excluded for having missing values in potential confounders. Thus, the analyses were conducted with 975 participants.”

Reviewer 2 Report

The study of Bayan-Bravo et al. is a longitudinal, population-based, observational investigation in a sample of 975 persons with age equal to or higher than 60 on the association with renal function decline of the habitual diet, as assessed by a validated food-frequency questionnaire.

Major points

The statistical analysis should be completed.

Regarding the dependent variable of the analysis, the authors should add validated renal endpoints such as eGFR change, or worsening of the CKD stage, or incidence of a given CKD stage. An analysis limited to a decrease of eGFR greater than the decrease expected by age is questionable for several reasons and impedes the comparison with other data.

Regarding the covariates, the authors should analyze additional multi-variable models including a key variable, that is the baseline eGFR. Baseline eGFR, by itself, is the strongest predictor of the change over time in kidney function in most of the long-term longitudinal studies. Accordingly, Table 1 should include descriptive data for baseline eGFR also.  In analyses on additional renal endpoints, the list of covariates should include follow-up duration also.

Author Response

Reviewer 2

The study of Bayan-Bravo et al. is a longitudinal, population-based, observational investigation in a sample of 975 persons with age equal to or higher than 60 on the association with renal function decline of the habitual diet, as assessed by a validated food-frequency questionnaire.

We appreciate the comments of this reviewer, which has certainly improved the manuscript. Following the recommendations of the reviewer, we have change the definition of renal function decline. In this version of the manuscript, you can see new tables 1, 2 and 3.

Major points

The statistical analysis should be completed.

Regarding the dependent variable of the analysis, the authors should add validated renal endpoints such as eGFR change, or worsening of the CKD stage, or incidence of a given CKD stage. An analysis limited to a decrease of eGFR greater than the decrease expected by age is questionable for several reasons and impedes the comparison with other data.

Authors: Thank you for your comment in this point.

We calculated incident CKD (those with an eGFR >60 at baseline and <60 in 2015) and only 35 participants developed incident CKD. When we explored the association with the Mediterranean diet, results were similar but more extreme. We decided not to show this model because it was based on few events.

To follow the recommendation of the reviewer we defined renal function decline as a decrease in eGFR greater than 1 mL/min/1.73m2 per year during follow-up. In this case, 104 events occurred. Accordingly, we have change tables 1, 2, and 3. We think this new definition of the end-point does not impede the comparison with other data.

It should be noted that this is a population-based sample (not selected for renal damage), and this is a primary prevention study conducted in the general population. In this population, the Mediterranean diet preserves renal function.

Here you can see the new version of tables 2 and 3 (in this new version all the models are adjusted for eGFR at baseline).

Table 2. Association between the adherence to the Mediterranean Diet Adherence Screener (MEDAS) index and the risk of renal function decline after 6.5-years of follow-up in the Seniors-ENRICA study (2008/10-2015) N=975.

Low adherence to MEDAS

(0-5 points)

OR (95% CI)

Moderate adherence to MEDAS

(6-8 points)

OR (95% CI)

High adherence to MEDAS

(9-14 points)

OR (95% CI)

p trend

Renal function decline

Cases/total

104/975

54/414

33/314

17/247

Model 1

Ref.

0.78 (0.49-1.26)

0.46 (0.26-0.83)†

0.009

Model 2

Ref.

0.67 (0.41-1.09)

0.53 (0.29-0.95)*

0.020

Model 3

Ref.

0.63 (0.38-1.03)

0.52 (0.29-0.95)*

0.015

Model 4

Ref.

0.61 (0.37-1.01)

0.52 (0.28-0.94)*

0.016

*p<0.05; † p<0.01 ;OR: Odds Ratio; CI: Confidence Interval.

Model 1: Adjusted for sex, age, eGFR at baseline, education level (primary, secondary, or university), and total energy intake.

Model 2: Additionally adjusted for smoking status (never, former, or current-smoker), alcohol consumption (habitual drinker, occasional drinker, non-drinker, or ex-drinker), leisure-time physical activity (MET-h/wk), household physical activity (MET-h/wk), occupational physical activity (sedentary, light, moderate/vigorous, or none), time spent watching TV (h/wk), night sleeping time (h/d), prevalence of cardiovascular disease, bronchitis or asthma, cancer, and osteo-muscular disease.

Model 3: Additionally adjusted for BMI at baseline and prevalent components of the metabolic syndrome (glucose ≥100 mg/dL (yes, no), blood pressure ≥130/85 mmHg (yes, no), abdominal obesity ≥ 102 cm in men and ≥ 88 cm in women (yes, no), HDL-c <40 mg/dL in men and <50 mg/dL in women (yes, no) , triglycerides ≥ 150 mg/dL (yes, no).

Model 4: : Additionally adjusted for an increase in BMI (yes, no) and incident components of the metabolic syndrome developed during 2008/10 to 2015 (glucose ≥100 mg/dL (yes, no), blood pressure ≥130/85 mmHg (yes, no), abdominal obesity ≥ 102 cm in men and ≥ 88 cm in women (yes, no), HDL-c <40 mg/dL in men and <50 mg/dL in women (yes, no), triglycerides ≥ 150 mg/dL (yes, no), and muscular mass (kg).

Table 3. Association between the adherence to a posteriori dietary patterns and the risk of renal function decline after 6.5-years of follow-up in the Seniors-ENRICA study (2008/10-2015) N=975.

Western-like pattern

Mediterranean-like pattern

Q1

OR (95% CI)

(Lowest)

Q2

OR (95% CI)

Q3

OR (95% CI)

Q4

OR (95% CI)

(Highest)

P

linear

trend

Q1

OR (95% CI)

(Lowest)

Q2

OR (95% CI)

Q3

OR (95% CI)

Q4

OR (95% CI)

(Highest)

P

linear

trend

Renal function decline

Cases/total

104/975

20/244

31/244

29/244

24/243

37/244

27/243

23/244

17/244

Model 1

1 Ref.

1.68 (0.91-3.09)

1.61 (0.84-3.07)

1.61 (0.76-3.42)

0.235

1 Ref.

0.68 (0.39-1.18)

0.58 (0.32-1.04)

0.40 (0.20-0.80)†

0.008

Model 2

1 Ref.

1.60 (0.85-3.00)

1.58 (0.81-3.08)

1.42 (0.65-3.09)

0.381

1 Ref.

0.67 (0.38-1.18)

0.63 (0.34-1.17)

0.45 (0.22-0.92)*

0.030

Model 3

1 Ref.

1.59 (0.85-2.98)

1.50 (0.77-2.92)

1.35 (0.61-2.97)

0.474

1 Ref.

0.67 (0.38-1.20)

0.65 (0.35-1.19)

0.47 (0.23-0.96)*

0.042

Model 4

1 Ref.

1.65 (0.87-3.12)

1.54 (0.78-3.05)

1.43 (0.64-3.18)

0.411

1 Ref.

0.70 (0.39-1.24)

0.68 (0.36-1.26)

0.47 (0.23-0.97)*

0.049

Model 5

1 Ref.

1.62 (0.86-3.07)

1.37 (0.68-2.73)

1.12 (0.49-2.57)

0.798

1 Ref.

0.68 (0.38-1.22)

0.66 (0.35-1.25)

0.45 (0.21-0.96)*

0.070

*p<0.05; † p<0.01. OR: Odds Ratio. CI: Confidence Interval.

Model 1: Adjusted for sex, age, eGFR at baseline, education level (primary, secondary, or university), and total energy intake.

Model 2: Additionally adjusted for smoking status (never, former, or current-smoker), alcohol consumption (habitual drinker, occasional drinker, non-drinker, or ex-drinker), leisure-time physical activity (MET-h/wk), household physical activity (MET-h/wk), occupational physical activity (sedentary, light, moderate/vigorous, or none), time spent watching TV (h/wk), night sleeping time (h/d), prevalence of cardiovascular disease, bronchitis or asthma, cancer, and osteo-muscular disease.

Model 3: Additionally adjusted for BMI at baseline and prevalet components of the metabolic syndrome (glucose ≥100 mg/dL (yes, no), blood pressure ≥130/85 mmHg (yes, no), abdominal obesity ≥ 102 cm in men and ≥ 88 cm in women (yes, no), HDL-c <40 mg/dL in men and <50 mg/dL in women (yes, no) , triglycerides ≥ 150 mg/dL (yes, no).

Model 4: Additionally adjusted for an increase in BMI (yes, no) and incident components of the metabolic syndrome developed during 2008/10 to 2015 (glucose ≥100 mg/dL (yes, no), blood pressure ≥130/85 mmHg (yes, no), abdominal obesity ≥ 102 cm in men and ≥ 88 cm in women (yes, no), HDL-c <40 mg/dL in men and <50 mg/dL in women (yes, no), triglycerides ≥ 150 mg/dL (yes, no), and muscular mass (kg).

Model 5: Additionally adjusted for the other “a posteriori” dietary pattern, as appropriate.

Regarding the covariates, the authors should analyze additional multi-variable models including a key variable, that is the baseline eGFR. Baseline eGFR, by itself, is the strongest predictor of the change over time in kidney function in most of the long-term longitudinal studies.

Authors: eGFR is now included as a covariate in all the models. We have modify all the tables accordingly. It is of note that new tables are very similar due to the high number of covariates (both prevalent and incident) which had already been considered. Please, see the new version of tables 2 and 3.

Accordingly, Table 1 should include descriptive data for baseline eGFR also.  

Authors: Now, you can see the description for baseline eGFR in the new version of table 1. We have modify all the results accordingly. We have also included the stratification according to eGFR categories (The Kidney Disease: Improving Global Outcomes KDIGO 2012).

Table 1. Baseline characteristics of the Seniors-ENRICA participants according to renal function decline after 6.5 years of follow-up (2008/10-2015) N=975.

Total

No renal function decline

Renal function decline

p

value

N (%)

975 (100)

871 (89.3)

104 (10.7)

Sex, women, %

48.6

49.7

39.4

0.047

Age, in years, mean (SD)

67.4 (5.43)

67.2 (5.3)

69.0 (6.1)

0.014

eGFR  (SD)

77.9 (12.2)

78.2 (12.0)

76.5 (14.2)

0.189

eGFR categories, %

Category 1 (≥90)

18.2

18.4

16.4

0.027

Category 2 (≥60-89)

74.2

74.4

72.1

Category 3a (≥45-59)

6.8

6.5

8.7

Category 3b (≥30-44)

0.8

0.7

1.9

Category 4 (≥15-29)

0.1

0.0

1.0

Category 5 (<15)

0

0

0

Serum creatinine

0.90 (0.18)

0.89 (0.17)

0.93 (0.20)

0.048

Educational level, %

Primary

48.0

47.3

53.9

0.339

Secondary

26.7

27.3

21.1

University

25.3

25.4

25.0

Energy intake in kcal, mean (SD)

2046 (571)

2054 (565)

1974 (617)

0.182

Smoking status, %

Never smoker

56.6

57.2

51.9

0.032

Former smoker

31.9

32.3

28.9

Current Smoker

11.5

10.6

19.2

Alcohol consumption, %

Habitual drinker

47.5

48.5

39.4

0.244

Occasional drinker

21.4

21.4

22.1

Non-drinker

23.6

22.7

30.8

Ex-drinker

7.5

7.5

7.7

Leisure-time physical activity, MET-h/wk, mean (SD)

22.6 (15.7)

22.9 (15.6)

21.0 (16.2)

0.075

Household physical activity, MET-h/wk, mean (SD)

36.8 (31.6)

37.5 (31.4)

31.0 (32.5)

0.049

Time spent watching TV in h/wk, mean (SD) 

17.4 (11.1)

17.0 (10.8)

20.1 (13.1)

0.008

Night sleeping time in h/d, mean (SD)

7.14 (1.43)

7.13 (1.43)

7.20 (1.49)

0.664

Cardiovascular disease,%  

1.23

0.92

3.85

0.010

Bronchitis or asthma, %

7.7

7.8

6.7

0.697

Cancer, %

2.4

2.4

1.9

0.096

Osteo-muscular disease, % 

46.7

46.0

51.9

0.256

BMI at baseline in kg/m2, mean (SD)

28.5 (4.16)

28.5 (4.1)

29.1 (4.6)

0.119

Metabolic syndrome components at baseline

Glucose ≥100, %

43.6

42.6

51.2

0.070

Blood pressure ≥130/85, %

80.7

80.6

81.7

0.782

WC ≥102 men, ≥ 88 women, %

56.3

55.3

64.4

0.078

HDL-c <40 men, <50 women, %

22.8

21.8

30.8

0.040

Triglycerides ≥150, %

20.9

19.9

29.8

0.018

BMI in 2015 in kg/m2, mean (SD)

27.9 (4.3)

27.9 (4.2)

28.6 (4.6)

0.073

Metabolic syndrome components in 2015

Glucose ≥100, %

41.0

40.0

50.0

0.049

Blood pressure ≥130/85, %

66.7

65.8

74.0

0.092

WC ≥102 men, ≥ 88 women, %

59.4

58.7

65.4

0.187

HDL-c <40 men, <50 women, %

27.9

26.4

40.4

0.003

Triglycerides ≥150, %

16.5

15.7

23.1

0.056

Muscle mass in kg, mean (SD)

46.7 (9.4)

46.6 (9.5)

47.9 (9.1)

0.197

SD (Standard Deviation); eGFR: estimated glomerular filtration rate (Stratification according to the Kidney Disease: Improving Global Outcomes KDIGO 2012). BMI: Body Mass Index, WC: waist circumference; HDL-c: High-density lipoprotein. *P values are based on chi-square test for qualitative variables or t-test for continuous variables.

In analyses on additional renal endpoints, the list of covariates should include follow-up duration also.

Authors: The new version includes a reference to the duration of follow-up for the calculation of incident covariates.

Round 2

Reviewer 2 Report

The authors addressed the points raised by this reviewer.

Author Response

Thank you for your insights and your comments that really improved the manuscript.